# Whole-Genome Sequencing and Biotechnological Potential Assessment of Two Bacterial Strains Isolated from Poultry Farms in Belgorod, Russia

**DOI:** 10.3390/microorganisms11092235

**Published:** 2023-09-05

**Authors:** Vladislav Yu. Senchenkov, Nikita S. Lyakhovchenko, Ilya A. Nikishin, Dmitry A. Myagkov, Anna A. Chepurina, Valentina N. Polivtseva, Tatiana N. Abashina, Yanina A. Delegan, Tatiana B. Nikulicheva, Ivan S. Nikulin, Alexander G. Bogun, Viktor I. Solomentsev, Inna P. Solyanikova

**Affiliations:** 1Department of Biochemistry, Medical Institute, Belgorod State National Research University, 308015 Belgorod, Russia; senchenkov@bsu.edu.ru (V.Y.S.); lyakhovchenko@bsu.edu.ru (N.S.L.); 2Department of Biotechnology and Microbiology, Institute of Pharmacy, Chemistry and Biology, Belgorod State National Research University, 308015 Belgorod, Russia; 3Department of Biology, Institute of Pharmacy, Chemistry and Biology, Belgorod State National Research University, 308015 Belgorod, Russia; 4Institute of Biochemistry and Physiology of Microorganisms, Federal Research Center «Pushchino Scientific Center for Biological Research of Russian Academy of Sciences» (FRC PSCBR RAS), 142290 Pushchino, Russiatnabashina@gmail.com (T.N.A.); mewgia@yandex.ru (Y.A.D.); 5State Research Center for Applied Microbiology and Biotechnology, 142279 Obolensk, Russia; bogun62@mail.ru (A.G.B.); solomentsev@obolensk.org (V.I.S.); 6Fund of Innovative Scientific Technologies, 308518 Belgorod, Russia; nikulicheva@bsu.edu.ru (T.B.N.); nikulin@bsu.edu.ru (I.S.N.); 7Regional Microbiological Center, Belgorod State National Research University, 308015 Belgorod, Russia

**Keywords:** *Peribacillus frigoritolerans*, *Bacillus subtilis*, whole genome sequencing, ammonification, nitrification, microbial relationships, poultry waste

## Abstract

Bacteria, designated as A1.1 and A1.2, were isolated from poultry waste based on the ability to form ammonia on LB nutrient medium. Whole genome sequencing identified the studied strains as *Peribacillus frigoritolerans* VKM B-3700D (A1.1) and *Bacillus subtilis* VKM B-3701D (A1.2) with genome sizes of 5462638 and 4158287 bp, respectively. In the genome of *B. subtilis* VKM B-3701D, gene clusters of secondary metabolites of bacillin, subtilisin, bacilisin, surfactin, bacilliacin, fengycin, sactipeptide, and ratipeptide (spore killing factor) with potential antimicrobial activity were identified. Clusters of coronimine and peninodin production genes were found in *P. frigoritolerans* VKM B-3700D. Information on coronimine in bacteria is extremely limited. The study of the individual properties of the strains showed that the cultures are capable of biosynthesis of a number of enzymes, including amylases. The *B. subtilis* VKM V-3701D inhibited the growth of bacterial test cultures and reduced the growth rate of the mold fungus *Aspergillus unguis* VKM F-1754 by 70% relative to the control. The antimicrobial activity of *P. frigoritolerans* VKM V-3700D was insignificant. At the same time, a mixture of cultures *P. frigoritolerans* VKM B-3700D/*B. subtilis* VKM B-3701D reduced the growth rate of *A. unguis* VKM F-1754 by 24.5%. It has been shown that strain A1.1 is able to use nitrogen compounds for assimilation processes. It can be assumed that *P. frigoritolerans* VKM V-3700D belongs to the group of nitrifying or denitrifying microorganisms, which may be important in developing methods for reducing nitrogen load and eutrophication.

## 1. Introduction

Demographic and economic growth trends in developing countries are increasing the demand for animal proteins, including poultry [1]. According to some estimates, global meat production will increase by 18% by 2028 [1]. At the same time, meat production waste is projected to increase. Thus, solid waste from poultry production, including poultry manure, can be the cause of many environmental consequences. Malgorzata J. Kacprzak et al. [2], reported that, on average, farms produce about 180 g of bird droppings. For example, a poultry farm with 10,000 chickens can generate about 1800 kg of fresh manure per day [2]. In three months, 162,000 kg of poultry droppings can be generated. These amounts can lead to the development of pathogenic microflora or an excess of organic matter in one place, since the litter is a rich breeding ground for Enterobacteriaceae (e.g., *Salmonella* sp., *Enterococcus* sp., and *Escherichia coli* itself). In addition, the use of antibiotics in livestock and poultry is widespread, which can lead to the development of antibiotic-resistant groups of microorganisms [3].

Thus, the problem of utilizing poultry manure is urgent, because large areas are used for its storage, and the period of its natural degradation is long. Moreover, these waste accumulations are a source of objectionable odor and various financial costs [4].

One of the ways to reduce the risk and negative impact of poultry waste is composting [1]. There is evidence that the application of compost to the soil can increase the growth parameters of plants and soil microorganisms by providing nitrogen compounds [5], since poultry manure as an organic fertilizer is characterized by a high content of nutrients, such as nitrogen and phosphorus [6]. However, the concentration of organic, including humic, substances in the compost depends on many factors, including the duration of storage. At the same time, compost is characterized by its own siltation and, when applied to soils, this may affect the composition and activity of soil microflora [5]. The scientific literature describes cases of correlation between the number of soil microorganisms and the amount of organic fertilizer applied to the soil [7].

Composting involves the biochemical action on the organic substrate by various macro- and microorganisms and is conditionally divided into three main stages: (1) mesophilic composting, in which readily soluble compounds decompose with the release of heat; (2) thermophilic composting, which occurs by heating the compost, and in which degradation of polysaccharides, proteins, and fats, together with the death of some plant seeds and pathogens occurs; (3) cooling or ripening–humification, stabilization of compost and attenuation of basic processes [8]. Microbial participation is carried out by groups of meso- and thermophilic aerobic microorganisms, which provide hydrolysis of polymeric substrates. In various sources, compost microorganisms are represented by genera such as *Stenotrophomonas* [9], *Alcaligenes* [10], *Pseudomonas* [11], *Klebsiella* [12], *Xanthomonas, Achromobacter* [13], and *Caulobacter* [14], and the predominance of bacilli or actinobacteria can be a quality indicator for a composting process [8]. Representatives of these genera can participate in increasing the release of nitrogen in ammonium forms, and in the suppression of various plant, animal, and human pathogens [15]. Some microorganisms have nitrifying activity [16].

To increase the activity of positive microflora in composting, there are a number of commercial inoculants containing consortia of some species of microorganisms. Among them are Effective Microorganism (EM) and Microbial Activator Super LDD 1 [8], and the biological preparation bioExpert for composting.

Thus, as a measure to reduce the microbial contamination of the resulting compost, we can use the additional introduction of active forms of antagonist microorganisms with additional useful physiological properties.

One of the areas of work is evaluating the application of proteolytic microorganisms (ammonifiers) for accelerated biodegradation of compost [4]. However, the intense release of ammonia can lead to a shift in the balance in the microbial trophic system toward representatives of nitrifying (oxidizing ammonia to nitrites and nitrates) and denitrifying groups [17]. This may be due to the possible loss of nitrogen due to the fact that the molecular forms (a product of denitrification activity) volatilize [18].

The purpose of this study was to find biotechnologically relevant microorganisms in poultry manure, determine their individual properties, and evaluate the possible role of strains in the bioconversion of nitrogen in composting.

## 2. Materials and Methods

### 2.1. Strain Isolation

Microorganisms from poultry waste were isolated by serial dilutions on nutrient agar LB (g/L): peptone-10; yeast extract-5; NaCl-10; agar-20 [19].

Pure cultures were obtained using the depleting streak method [19]; after one day, individual colonies were passed onto cups with LB agar by microbiological streak.

The cultures required for further studies were selected on the basis of the highest deamination activity; hence, for qualitative accounting of alkaline hydrolyzed nitrogen, a modification of nutrient medium LB (g/L) was prepared: peptone-10; yeast extract-5; NaCl-10; agar-20; bromothymol blue indicator—0.1 mL 0.5% solution [20].

The seeding on this agar was performed by the microbiological streak method. The presence of ammonium nitrogen could be judged by the recovery of the indicator around the colony, which turned blue. The most productive colonies were those in which the average radius of the indicator color change zones was the largest.

The ability to form deaminases in isolates was evaluated by the ability to form ammonia on nutrient medium with peptone using the Nessler reagent. The presence of phenylpyruvate (a derivative of L-phenylalanine formed by deamination) in the nutrient medium was judged by the appearance of red-brown staining of the culture liquid when 2 mL of 10% Fe (III) chloride solution was added.

### 2.2. Cultural, Morphological, and Biochemical Characteristics

Cultural, morphological, and tinctorial properties were studied using standard methods [21].

The ratio to oxygen was determined by inoculation in an agar column. For this purpose, 3% peptone agar was poured into 8 mL tubes. After autoclaving, we sowed by injection into the solidified agar. The nature of growth in the thickness of the medium was used to judge the attitude of the culture toward O_2_ [21].

To determine nitrate reductase activity, the isolates were cultured in liquid nutrient medium (3% peptone) for 24 h and 1 mL of 0.1% KNO_3_ was added. After 1 h of incubation at 37 °C, 0.5 mL of Griess reagent was added to the test tube. In the case of a positive reaction, the medium turns red [22].

Proteolytic activity was evaluated by the ability of isolates to hydrolyze gelatin (12%), casein, and albumin. For this purpose, isolates were transferred into liquid mineral nutrient medium with proteins as a growth substrate.

Amylolytic activity was evaluated by the area of starch hydrolysis in Petri dishes (nutrient medium composition (g/L): soluble starch-10; peptone-10; KH_2_PO_4_-5; microbiological agar-20) after addition of Lugol solution. The cultures were incubated for 3 days.

The ability of isolates to use amino acids as a growth substrate was determined by grazing the strains in a liquid mineral medium (composition (g/L): KNO_3_-4; KH_2_PO_4_-0.6; Na_2_HPO_4_-1.4; MgSO_4_-0.8) containing amino acids at a concentration of 0.7 g/L.

Phosphate-solubilizing activity was evaluated by growth pattern on National Botanical Research Institute’s phosphate growth medium (NBRIP), consisting of (g/L) glucose-10; MgCl_2_-5; MgSO_4_-0.25; KCl-0.2; (NH_4_)_2_SO_4_-0.1; Ca_3_(PO)_4_-5; microbiological agar-2% [23]. The cultures were incubated at 25 °C for 3 days. Phosphate-solubilizing activity was judged by the formation of a lumen of dissolved phosphate at the edge of the culture in a Petri dish.

The ability to use carbohydrates as a growth substrate was evaluated on solid mineral nutrient medium containing D-glucose, sucrose, lactose, fructose, sodium benzoate, maltose, sorbitol, and mannitol at concentrations of 0.5%.

The minimum and maximum growth temperatures were determined by stippling on the nutrient medium (3% peptone agar) and incubated at 10, 20, 30, 40, and 50 °C. The relation of cultures to the temperature range was evaluated according to the growth pattern.

The minimum and maximum pH values for growth were determined by passage of bacterial suspension in the nutrient medium (3% liquid peptone) and incubated at pH 1 to 9. The ratio of cultures to the range in pH values was assessed by the growth pattern.

### 2.3. Resistance to Antibiotics

Antibiotic resistance of isolates was studied using the disc diffusion method. Eighty antibiotic discs (NICP, St. Petersburg, Russia) were used in the study (Table 1). To analyze the antibiotic resistance of isolates, 100 μL of culture in the exponential growth stage was applied to Petri dishes with solid LB medium and evenly distributed over the entire surface of the dish using a spatula. The antibiotic discs were placed on top of the culture at an equal distance from each other and from the edges of the Petri dishes. The results were evaluated after 24 h of cultivation at 25 °C by the presence of growth inhibition zones around the disc. If there was no such zone, the strain was resistant to the given concentration of antibiotic.

Selected isolates were compared to each other for some traits (from 0 to 1, where 0 is complete dissimilarity and 1 is complete similarity) using a matching coefficient S [19]:(1)$$S=\frac{a+d}{a+b+c+d}$$
where a + d is the sum of traits for which cultures A and B coincide (a—both strains are positive, d—both are negative), b is the sum of traits for which strain A is positive, B is negative, c is the sum of traits for which strain A is negative, strain B is positive.

### 2.4. Antagonistic Activity

In order to evaluate the antibacterial activity of isolated bacterial strains, the formation of the lysis zone of test cultures was also studied. For this study, new isolates were pre-grown in liquid LB medium and the bacterial suspension in the exponential growth phase was directly taken to study the antagonistic activity. One hundred micrograms of test culture in the exponential growth stage was applied to Petri dishes with solid LB medium and evenly distributed with a spatula over the entire surface of the dish. Sterile filter paper discs were placed on top of the test culture and 5 mcg of bacterial suspension was applied. The results were evaluated after 24 h of cultivation at 25 °C by the presence of a lysis zone around the paper disc. The presence of this zone indicates the activity of the new isolates against the test culture.

The antagonistic activity of isolated bacterial strains against such test cultures as *Janthinobacterium lividum* VKM B-3705D, *Bipolaris sorokiniana* VKM F-4006, *Alternaria brassicicola* VKM F-1864, *Pythium vexans* VKM F-1193, and *Aspergillus unguis* VKM F-1754 was estimated by perpendicular strokes [19]. The cultures were incubated at 25 °C.

The reliability of the difference between the averaged values was calculated statistically using the difference method [24,25].

### 2.5. Syntrophic Interactions of Strains

Syntrophic interactions of isolates and ammonia-oxidizing capacity during nutrient substrate conversion were evaluated by culturing strains in tanks with nutrient media, where the ammonia-forming bacterium in 3% liquid peptone was grown in a static surface way with aeration (400 mL/h), and air containing the forming gas was fed into the liquid mineral nutrient medium in a second tank without nitrogen with the companion strain (Figure 1).

The cultures were incubated for 5 days. Bacterial growth in the mineral nutrient medium was measured using a spectrophotometer (Nabi NB1-D-210108, Micro Digital, Seongnam-si, Republic of Korea) at wavelength λ = 600 nm. Nitrite content in the medium was estimated spectrophotometrically with Griess reagent (VECTON, Novosibirsk, Russia) in an empirically selected wavelength range λ = 800−1100 nm. The presence of ammonium forms of nitrogen was assessed using Nessler reagent (Urals Chemical Plant, LLC, Verkhnyaya Pyshma, Russia) at λ = 430 nm. The tanks without the ammonia-forming strain in 3% peptone liquid medium were used as a control variant).

### 2.6. Whole Genome Sequencing

Genomic DNA of strains A1.1 and A1.2 was isolated from fresh culture biomass grown from 1 colony using the QIAamp DNA Mini Kit (cat. #51304; Qiagen, Hilden, Germany). The SG GM full-genome library preparation kit (Raissol Bio, Moscow Region, Russia) was used to prepare libraries. Sequencing was performed using GenolabM equipment.

Quality control of the reads was performed using HTQC [26]. Low-quality (Q < 10), short (<100 bp) reads and adapter sequences were deleted using Trimmomatic software (version 0.38) [27]. The quality control results of the reads are presented in Table 2. Quality control removed 9.88% (A1.1) and 13.38% (A1.2) of the total number of raw reads.

Raw filtered reads were collected using SPAdes software version 3.15.4 [28] at k-mer sizes 57 (A1.1) and 95 (A1.2). Contigs shorter than 500 bp were removed. The metrics of the assemblies are presented in Table 3.

Genome structural annotation and genome search for target genes were performed using Prokka [29], RAST [30], and NCBI Prokaryotic Genome Annotation Pipeline (PGAP) software and web services (https://www.ncbi.nlm.nih.gov/genome/annotation_prok/, accessed on May 2023). Functional annotation was performed using BlastKoala ver. 3.0 (https://www.kegg.jp/blastkoala/, accessed on May 2023) [31], and KEGG (https://www.genome.jp/kegg/, release 2023/04, accessed on May 2023) [32]. ANI (https://www.ezbiocloud.net/tools/ani, accessed on May 2023) [33] and DDH [34] parameters were used to identify strains. The sequences of the strains were deposited in Genbank database under the following numbers:

A1.1.–Bioproject PRJNA951603, Biosample SAMN34045332, Genbank acc number JARTEM000000000

A1.2.–Bioproject PRJNA951604, Biosample SAMN34045556, Genbank acc number JARTEN000000000

## 3. Results

### 3.1. Strains and Whole Genome Sequencing

After the initial isolation and purification of the isolates, more than 20 microorganisms were found to be present in the sample.

A subsequent test using the bromothymol blue indicator showed that only 2 isolates had deamination activity: A1.1 and A1.2. Further, only these microorganisms participated in the study.

According to BLAST analysis of the *gyrB* genes, strain A1.1 could belong to *Peribacillus frigoritolerans*, *Peribacillus simplex*, or *Peribacillus muralis*; strain A1.2 could belong to *Bacillus subtilis*. We compared the complete genomes of the strains under study with the complete genomes of the type strains of the most closely related species (Table 4).

Thus, we can identify strain A1.1 as *Peribacillus frigoritolerans* and strain A1.2 as *Bacillus subtilis*.

The cultures were deposited in the All-Russian collection of microorganisms as *Peribacillus frigoritolerans* VKM B-3700D and *Bacillus subtilis* VKM B-3701D.

Strain A1.2 (Figure 2a) is a typical representative of its species group (*B. subtilis*). Gene clusters of secondary metabolites production characteristic of *B. subtilis* were found in its genome, namely bacillin, subtilisin, bacilisin, surfactin, bacillibactin, fengicin, sactipeptide, and ratipeptide (spore killing factor). Most of the compounds that this strain is theoretically capable of producing are known for their antimicrobial activity. Bacillin is a polyene antibiotic that was first isolated from the culture fluid of *Bacillus subtilis* and showed antibiotic activity against some of the test cultures used by the authors during the study [35].

Sactipeptides exhibit various biological activities, such as antibacterial, spermicidal, and hemolytic properties [36]. For example, Hudson et al. [37] described a new sactipeptide, huazacin, with growth-suppressing activity against *Listeria monocytogenes*. Rantipeptides (formerly known as “SCIFF peptides”) are ribosome synthesized and post-translationally modified peptides (RiPP) common among members of the *Clostridia* class [38,39,40]. There is evidence that these compounds play an essential role in the regulation of population abundance through the quorum-sensing mechanism [41].

In the *P. frigoritolerans* strain VKM B-3700D (Figure 2a), it is interesting to note the presence of koranimine and paeninodine gene production clusters. Not much is known about koranimine. According to Evans et al. [42], koranimine produced by a representative of *Bacillus* sp. had no antibiotic effect against *Escherichia coli*, *Staphylococcus aureus*, *Enterococcus faecalis*, *Klebsiella pneumoniae*, or *Saccharomyces cerevisiae*.

Paeninodine is a peptide first isolated from *Paenibacillus dendritiformis* [43]. Zhu et al. tested it for antimicrobial activity against the following targets: *B. subtilis*, *M. flavus*, *B. rhizoxinica*, *S. japonicum*, *X. citri pv. mangiferaeindicae*, *B. cereus*, *B. megaterium*, and *B. amyloliquefaciens*, but no antimicrobial properties were found for paeninodine.

### 3.2. Physiological and Biochemical Characteristics of the Strain

The bacteria *P. frigoritolerans* VKM B-3700D and *B. subtilis* VKM B-3701D are bacilliform, Gram-positive, spore-forming bacteria. The *P. frigoritolerans* strain VKM B-3700D is motile, whereas *B. subtilis* VKM B-3701D is not (Table 5). The isolates appeared to be facultative anaerobes. Both isolates tested positive for nitrate reductase, oxidase, and catalase activities (Table 6).

During the evaluation of the proteolytic activity of the cultures, it was found that both isolates were capable of hydrolyzing gelatin and albumin and only *B. subtilis* VKM B-3701D was capable of using casein as a growth substrate (Table 6).

Isolates form ammonia when cultured on 3% peptone medium. *P. frigoritolerans* VKM B-3700D was capable of using D-glucose, sucrose, maltose, mannitol and sorbitol, cysteine, phenylalanine, dehydroxyphenylalanine, isoleucine, glutamine, and ornithine, but not fructose and lactose, histidine, tyrosine, threonine, serine, norleucine, or lysine. In turn, *B. subtilis* VKM B-3701D uses D-glucose, sucrose, lactose, maltose, mannitol, sorbitol, tyrosine, phenylalanine, dihydroxyphenylalanine, isoleucine, and lysine, and does not use fructose, histidine, cysteine, threonine, serine, norleucine, or glutamine, ornithine. None of the strains grow on sodium benzoate as a growth substrate (Table 6).

The positive test reaction for nitrate-reducing activity (Table 6) suggests that the strains may participate in aerobic nitrite formation [44], and the isolated strains may participate in nitrogen conversion [45]. By converting nitrates into nitrites, the cultures provide plants with an available form of nitrogen. This property may be promising in agrobiotechnology. For example, microorganisms are often used as components of biopreparations that can intensify nitrogen fixation and increase yields [46]. At the same time, the urease activity test was negative for the isolates. Consequently, they cannot extract ammonium from urea.

The lipolytic activity of *B. subtilis* strain A1.2 (Table 6) suggests that the culture can be a biotechnologically valuable producer of lipases, which can be used for the production of pharmaceuticals, cosmetics, detergents, as a food additive, in perfumery, etc. [47]. Prospective use can be evaluated by determining the enzymatic activity depending on cultivation conditions.

The study shows that A1.2 can belong to the group of phosphate-solubilizing bacteria due to its ability to dissolve calcium phosphate (Ca_3_(PO_4_)_3_). It is known that microorganisms of this group can be used as a biotechnological solution to phosphate deficiency in soil, as they contribute to its conversion into a form accessible to plants [48].

In addition, the biotechnological significance of the isolated cultures is determined by the proteolytic activity and the ability to form ammonia (Table 6). Proteolytic enzymes are used in food production, detergents, bioremediation technologies, and the production of biologically active peptides. The pharmaceutical industry uses collagenases, for example, to treat burns. Whey protein produced in the production of cheese is known to have nutritional value due to its protein content. Therefore, hydrolysis of these proteins is performed by acidic and alkaline peptidases, which allows them to be used in the production of food for children with intolerance to milk proteins [49].

### 3.3. Antibiotic Resistance

The isolates isolated are sensitive to most of the 80 studied antibiotics. The *P. frigoritolerans* strain VKM B-3700D is resistant to antibiotics such as amphotericin B (40 µg), bacitracin (0.04 units), bile (sodium deoxycholate) (3 µg), ketoconazole (20 µg), optoquine (6 µg), and saponin (750 µg). *B. subtilis* strain VKM B-3701D is resistant to antibiotics such as aztreonam (30 µg), amphotericin B (40 µg), bacitracin (0.04 units), and nystatin (80 units).

### 3.4. Antagonistic Activity

A study of the antagonistic activity of the new isolates by the test-culture lysis zone assessment method showed that both strains were inactive against all Gram-negative test cultures, except for *J. lividum* VKM B-3705D. Additionally, *P. frigoritolerans* VKM B-3700D and *B. subtilis* strains VKM B-3701D showed antagonistic activity against a number of Gram-positive test cultures (Table 7).

The evaluation of the antagonistic activity of the isolates revealed that *B. subtilis* VKM B-3701D had a significant effect in suppressing the growth of the culture, *J. lividum* VKM B-3705D inoculated with shade (Figure 3). In turn, strain A1.1 was ineffective against the test culture.

During the assessment of the antifungal activity of the isolates against *A. unguis* VKM F-1754, *B. sorokiniana* VKM F-4006, *A. brassicicola* VKM F-1864, and *P. vexans* VKM F-1193, it was revealed that strain *B. subtillis* VKM B-3701D showed antagonistic activity in all cases, whereas *P. frigoritolerans* VKM B-3700D was inactive only against *B. sorokiniana* VKM F-4006, and the other test cultures were suppressed.

The coefficient of concordance (S) calculated on the basis of the individual properties of the strains was 0.68. We can assume that the strains are close to each other by taxonomic affiliation.

### 3.5. Syntrophic Interactions of Strains

The mixture of two strains represents a community of two populations, presumably of the same species, which may be characterized by the “product–substrate” syntrophy-type interaction. Based on the fact that the isolates were selected on the basis of ammonia formation, it was found that *P. frigoritolerans* strain VKM B-3700D was capable of using nitrogen compounds for assimilation processes, as the specific growth of the culture on medium without nitrogen, in which air from the cultivated tank with *B. subtilis* VKM B-3701D served as the source of ammonium, exceeded the control version (blown with air from a sterile tank) by 90% (Figure 4).

Furthermore, in the culture liquid of *P. frigoritolerans* VKM B-3700D purged with air from a container with *B. subtilis* VKM B-3701D, ammonium (reaction with Nessler’s reagent) and nitrite (reaction with Griss reagent) were detected, while when purged with air from a sterile container, reactions for ammonia detection in *P. frigoritolerans* VKM B-3700D medium were negative. Thus, it can be assumed that isolate A1.1 oxidizes incoming ammonium into nitrite and nitrate due to the activity of nitrate reductases.

## 4. Discussion

Based on the fact that the violacein formed by the *J. lividum* VKM B-3705D is associated with the quorum-sensing mechanism [50], the suppression of pigmentation of the test culture is of interest because the mechanism is related to the virulence of many plant, animal, and human pathogenic microorganisms [17]. In addition, the antifungal activity of the strains represents agrobiotechnological potential in the field of plant protection.

Full-genome sequencing revealed that *P. frigoritolerans* VKM B-3700D and *B. subtilis* VKM B-3701D strains are potential producers of a number of biologically active compounds. Among them is koranimine. Little is known about this compound, however, it is assumed to have nematicidal activity against the wood lesion pathogen *Bursaphelenchus xylophilus* [51].

Since the isolates were isolated in a mixture from a single source, it was assumed that they were functionally related, which was shown when evaluating the ability of the *P. frigoritolerans* VKM B-3700D to use air from *B. subtilis* VKM B-3701D containing ammonium nitrogen. Thus, nitrogen compounds formed by *B. subtilis* VKM B-3701D strain promote growth of *P. frigoritolerans* VKM B-3700D companion culture and it can be assumed that the system may be characterized by the “product–substrate” component (isolate) ratio, or syntrophy.

It is known that the oxidation of ammonia to nitrite or nitrates is carried out in nature by nitrifying microorganisms. The resulting nitrogen compounds are reduced to oxides and free molecular nitrogen by denitrifying representatives [51]. Thus, the isolated strain *P. frigoritolerans* VKM B-3700D can refer to nitrifier or denitrifier.

Bacteria capable of removing nitrogen from media can be promising in nitrogen loading of soils and in preventing the development of eutrophication, a global problem that is a consequence of large amounts of nitrogen and phosphorus, manifesting itself in intensive development of algae (“blooming” of water bodies). Eutrophication can be caused by toxicogenic autotrophs, which negatively affects aquatic organisms and the aesthetic parameters of recreational areas due to objectionable odor [52].

## 5. Conclusions

Thus, Gram-positive, bacilliform, facultatively anaerobic bacteria designated as *P. frigoritolerans* VKM B-3700D (A1.1) and *B. subtilis* VKM B-3701D (A1.2) were isolated from poultry waste. During evaluation, their individual properties showed that the isolates had biotechnological potential, since the arsenal of enzymatic activities included both the formation of significant enzymes and the activity of strains that could be of interest. Thus, strain A1.1 was characterized by significant antagonistic, ammonifying, and phosphate-solubilizing activity, and the ability to synthesize amylases may be promising for producing enzymes for the food industry. In turn, the A1.1 strain was found to be capable of oxidizing ammonium nitrogen to nitrates and nitrites, which may be promising in reducing the nitrogen load of soils and preventing eutrophication.

In the scientific literature, cases of pathogenicity of *B. subtilis* and *P. frigoritolerans* bacteria have not been identified over the past 60 years. The isolated strains are susceptible to most antibiotics, but there are no specific threshold concentrations to assess the resistance profile. In the future, the isolated strains can be used in industry as they are not pathogenic. Moreover, at present, strains of *B. subtilis* are widely used in the composition of biological products, for example, for plant protection.

Further research involves in-depth study of the mechanisms of metabolic activity realization and the selection of regimes for obtaining the target products using *P. frigoritolerans* VKM B-3700D and *B. subtilis* VKM B-3701D cultures, together with the evaluation of the biotechnological process development. It is necessary to study the growth characteristics of the studied crops depending on the pH and the concentrations of salts, ammonium ions, and nitrates, since the data presented correspond only to optimal temperature conditions. In addition, research directions are needed concerning secondary metabolite potential of the strains for their significance in public health and veterinary and plant protection.

## Figures and Tables

**Figure 1 microorganisms-11-02235-f001:**
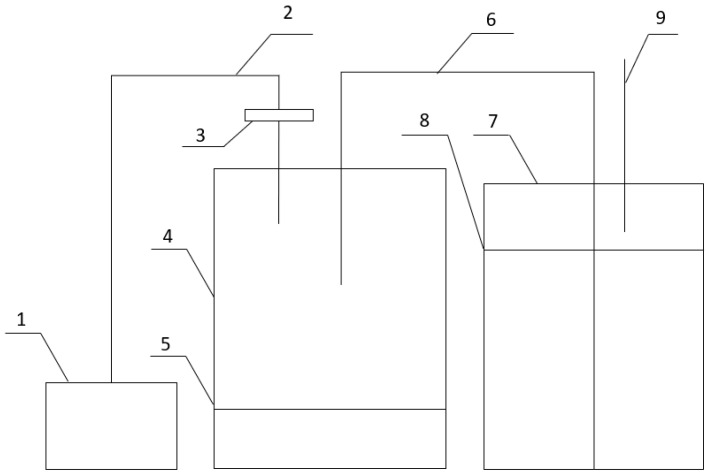
General scheme showing the installation to determine the syntrophic relationships of microorganisms: 1—compressor; 2—air supply tube; 3—air filter with a pore diameter of 0.22 μm; 4, 5—tank for cultivation and nutrient medium for surface cultivation of ammonia-forming bacteria; 6—tube for draining air containing ammonia; 7, 8—tank for cultivation and growth medium for deep cultivation of ammonia-oxidizing bacteria; 9—tube for excess air outlet.

**Figure 2 microorganisms-11-02235-f002:**
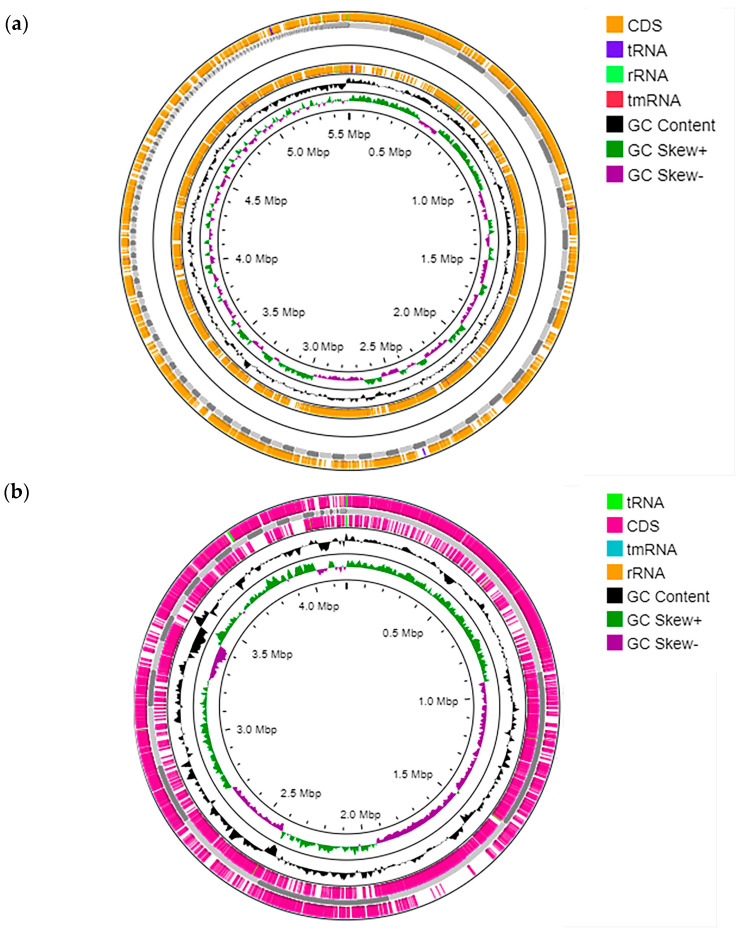
Circle maps of the genomes of the strains *P. frigoritolerans* VKM B-3700D (A1.1) (**a**) and *B. subtilis* VKM B-3700D (A1.2) (**b**). From outside to the center: all CDS and RNA genes on forward strand, all CDS and RNA genes on reverse strand, GC content, and GC skew.

**Figure 3 microorganisms-11-02235-f003:**
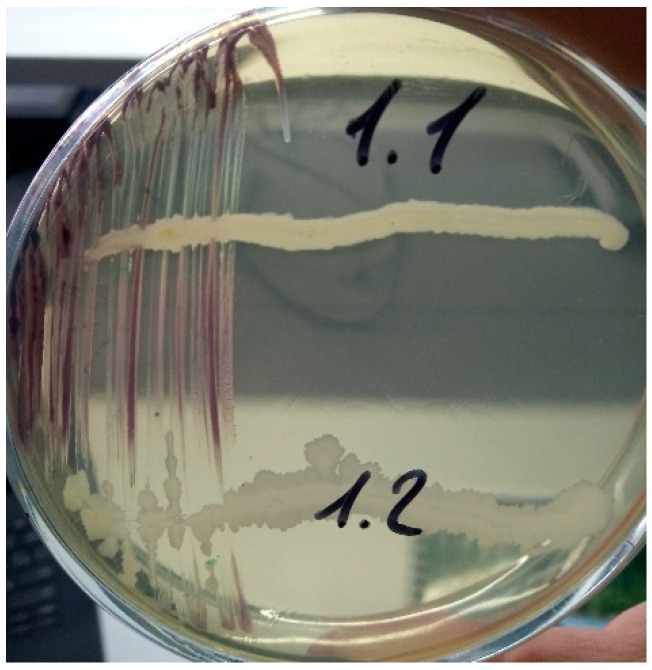
Suppression of *J. lividum* test culture VKM B-3705D by *P. frigoritolerans* VKM B-3700D and *B. subtilis* VKM B-3700D.

**Figure 4 microorganisms-11-02235-f004:**
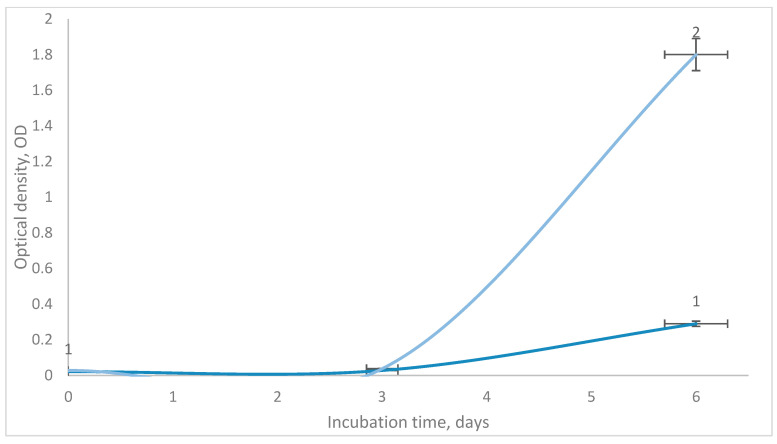
Changes in the optical density of *P. frigoritolerans* VKM B-3700D culture on medium without nitrogen (1), and in the variant where the source of ammonium was air from a tank with *B. subtilis* VKM B-3701D containing ammonium (2).

**Table 1 microorganisms-11-02235-t001:** Antibiotics used and their concentrations for assessing antibiotic resistance of isolates.

№	Antibiotic	Concentration	№	Antibiotic	Concentration
1	Azithromycin	15 μg	41	Oleandomycin	15 μg
2	Azlocilin	75 μg	42	Optochin	6 μg
3	Aztreonam	30 μg	43	Ofloxacin	5 μg
4	Amikacin	30 μg	44	Pefloxacin	5 μg
5	Amoscicillin	20 μg	45	Piperacillin	100 μg
6	Amoscyillin/clavulanic acid	20/10 μg	46	Polymxin	300 units
7	Ampicillin	10 μg	47	Rifampicin	5 μg
8	Ampicillin/sulbactam	10/10 μg	48	Roxithromycin	30 μg
9	Amphotericin B	40 μg	49	Saponin	750 μg
10	Bacitracin	10 units	50	Sisomycin	10 μg
11	Bacitracin	0.04 units	51	Sparfloxacin	5 μg
12	Benzylpenicillin	10 units	52	Streptomycin	300 μg
13	Vancomycin	30 μg	53	Sulfonamide	300 μg
14	Gentamicin	10 μg	54	Tetracycline	30 μg
15	Gentamicin	120 μg	55	Tylosin	30 μg
16	Doxycycline	30 μg	56	Tobramycin	10 μg
17	Bile (sodium deoxycholate)	3 mlg	57	Trimethoprim/sulfamethoxazole	1,25/23,75 μg
18	Imipenem	10 μg	58	Fluconazole	40 μg
19	Itraconazole	10 μg	59	Fosfomycin	200 μg
20	Kanamycin	30 μg	60	Fusidic acid	10 μg
21	Carbenicillin	25 μg	61	Furagin	300 μg
22	Carbenicillin	100 μg	62	Furadonin	300 μg
23	Ketoconazole	20 μg	63	Furazolidone	300 μg
24	Clarithromycin	15 μg	64	Cefazolin	30 μg
25	Clindamycin	2 μg	65	Cefaclor	30 μg
26	Clotrimazole	10 μg	66	Cephalexin	30 μg
27	Levomycin	30 μg	67	Cephalotin	30 μg
28	Levofloxacin	5 μg	68	Cefepim	30 μg
29	Linezolid	30 μg	69	Cefixime	5 mg
30	Lincomycin	15 μg	70	Cefoxitin	30 μg
31	Lomefloxacin	10 μg	71	Cefoperazone	75 μg
32	Meropenem	10 μg	72	Cefoperazone/sulbactam	50/50 μg
33	Moxiflocacin	5 μg	73	Cefotaxime	30 μg
34	Nalidixic acid	30 μg	74	Ceftazidime	30 μg
35	Neomycin	30 μg	75	Ceftriaxone	30 μg
36	Nystatin	80 ed	76	Cefuroxime	30 μg
37	Novobiocin	5 μg	77	Ciprofloxacin	5 μg
38	Norfloxacin	10 μg	78	Enrofloxacin	5 μg
39	Oxacillin	10 μg	79	Erythromycin	15 μg
40	Oxacillin	1 μg	80	Ertapenem	10 μg

**Table 2 microorganisms-11-02235-t002:** Results of quality control of readings, performed with HTQC [26].

Sample		Before QC	After QC
A1.1	Length of readings, bp	50–150	100–150
	Number of readings	2 × 788,236	2 × 53,208 (67.54%), and also Forward Only 164,997 (20.93%) Reverse Only 12,982 (1.65%)
A1.2	Length of readings, bp	50–150	100–150
	Number of readings	2 × 1,181,751	2 × 713,575 (60.38%), and also Forward Only 290,599 (24.59%) Reverse Only 19,508 (1.65%)

**Table 3 microorganisms-11-02235-t003:** Strain genome assembly metric A1.1 and A1.2.

	A1.1 (k = 57)	A1.2 (k = 95)
Genome length, bp	5,462,638	4,158,287
Number of contigs	377	72
N50, bp	34,168	351,311
N75, bp	16,020	71,083
N90, bp	6810	42,968
The shortest contig, bp	503	521
The longest contig, bp	161,241	542,378

**Table 4 microorganisms-11-02235-t004:** Results of a comparison of the complete genomes of the studied strains with typical closely related strains.

Sample	Relative Strains	ANI Value, %	DDH, %
A1.1	*Peribacillus frigoritolerans* DSM8801T (JALJWT000000000.1)	97.24	85.50
	*Peribacillus simplex* NBRC15720T (NZ_CP017704.1)	93.39	68.20
	*Peribacillus muralis* DSM16288T (LMBV01.1)	84.70	35.00
A1.2	*Bacillus subtilis* DSM10T (JAEPVU01.1)	100.00	100.00

**Table 5 microorganisms-11-02235-t005:** Cultural and morphological properties of isolates of *Peribacillus frigoritolerans* VKM B-3700D and *Bacillus subtilis* VKM B-3701D.

Properties	Cultures	
	*Peribacillus frigoritolerans* VKM B-3700D	*Bacillus subtilis* VKM B-3701D
Colony shape	Round	Incorrect
Colony color	Cream	White
Cell shape	Short sticks	Sticks
Gram stain reaction	+	+
Motility	+	-
Spore formation	+	+

Note: «+»—property availability; «-»—no property.

**Table 6 microorganisms-11-02235-t006:** Physiological and biochemical properties of isolates of *P. frigoritolerans* VKM B-3700D and *Bacillus subtilis* VKM B-3701D.

Properties	Cultures	
	*Peribacillus frigoritolerans* VKM B-3700D	*Bacillus subtilis* VKM B-3701D
Relation to oxygen	facultative anaerobe	facultative anaerobe
Temperature range (optimal)	20–40 °C (30 °C)	10–40 °C (40 °C)
pH range (optimal)	2–9 (7)	4–9 (7)
NaCl concentration range	0–6%	0–9%
Oxidase activity	+	+
Nitrate-reductase activity	+	+
Catalase activity	+	+
Urease activity	-	-
Lipolytic activity	-	+
Lecithinase activity	-	+
Amylolytic activity	-	+
Ability to dissolve inorganic phosphate	-	+
Ability to hydrolyze gelatin	+	+
Casein hydrolysis capacity	-	+
Ability to use albumin	+	+
Ability to hydrolyze Na-carboxymethylcellulose	-	-
Indole formation	-	-
Ammonia formation during growth on peptone (deamination activity)	+	+
Formation H_2_S	-	-
Growth on medium containing as the only substrate:		
D-glucose	+	+
Sucrose	+	+
Fructose	-	-
Lactose	-	+
Maltose	+	+
Mannitol	+	+
Sorbitol	+	+
Sodium benzoate	-	-
Histidine	-	-
Tyrosine	-	+
DL-Cysteine	+	-
DL-threonine	-	-
DL-phenylalanine	+	+
Dehydroxyphenylalanine	+	+
Isoleucine	+	+
Serin	-	-
Norleucine	-	-
Glutamine	+	-
Ornithine	+	-
Lysine	-	+

Note: «+»—property availability; «-»—no property.

**Table 7 microorganisms-11-02235-t007:** Antagonistic activity of new isolates.

Bacteria Test	Presence of Lysis Zone (Growth Suppression)	
	*P. frigoritolerans* VKM B-3700D	*Bacillus subtilis* VKM B-3701D
Gram-negative bacteria		
*Alcaligenes faecalis* B1518	-	-
*Echerichia coli* C600	-	-
*Pantoea agglomerans* ATCC 27155	-	-
*Pectobacterium carotovorum* B15	-	-
*Pseudomonas putida* KT2442	-	-
*P. aeruginosa* ML4262	-	-
*P. protegens* 38a	-	-
*Janthinobacterium lividum* BKM B-3705D	-	+
Gram-positive bacteria		
*Arthrobacter* sp. B52	-	-
*Bacillus cereus* GA5T	-	-
*B. subtilis* ATCC 6633	-	-
*B. thuringiensis* ATCC 35646	-	+
*Lysinibacillus sphaericus* VKM B-509	+	-
*Micrococcus luteus* VKM Ac-2230	+	+
*Kocuria rosea* VKM B-1236	-	+
*Staphylococcus aureus* St35	+	+
*Aeromonas veronii*	+	+
*Deinococcus radiodurans*	+	+
Mold fungi		
*Bipolaris sorokiniana* VKM F-4006	-	+
*Alternaria brassicicola* VKM F-1864	+	+
*Pythium vexans* VKM F-1193	+	+
*Aspergillus unguis* VKM F-1754	+	+

Note: «+»—property availability; «-»—no property.

## Data Availability

Data are contained within the article.

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
