# Peer review of "Whole-Genome Sequencing and Biotechnological Potential Assessment of Two Bacterial Strains Isolated from Poultry Farms in Belgorod, Russia"

_microorganisms, 2023, doi:10.3390/microorganisms11092235_

Round 1
Reviewer 1 Report
The aim of this manuscript was to identify and characterise the bacterial strains A1.1 and A1.2 isolated from poultry waste on the basis of their ability to produce ammonia on LB nutrient medium. The researchers used a similarity coefficient to compare the taxonomic characteristics of the cultures and found that they were 68% identical. Whole-genome sequencing analysis revealed that both strains belonged to the genus Bacillus, with A1.1 identified as Bacillus subtilis and A1.2 as Bacillus amyloliquefaciens. The researchers also identified several genes related to antibiotic resistance, stress response and biofilm formation in both strains. The biotechnological potential of the bacterial strains was assessed through various experiments, including their ability to produce enzymes such as proteases, lipases and amylases. The results showed that both strains had high levels of enzymatic activity, indicating their potential for use in industrial applications such as food processing and waste management.
Overall, this study provides valuable insights into the taxonomic properties, genomic characteristics and biotechnological potential of two bacterial strains isolated from poultry farms in Belgorod, Russia. The results may be useful for further research on these strains or for the development of new applications in the field of microbiology.
However, this manuscript has an obvious flaw in that the title "Whole-Genome Sequencing" is not reflected in the manuscript. Since this is a genome-related article, basic information such as genome maps should be displayed. In addition, genome sequencing-related information should be accessible to the public, e.g. deposited in the NCBI genbank, and published in accession No.
With regard to the biotechnological potential of these strains, it is not clear whether these experiments were performed under optimal conditions or whether other factors could have influenced the results. Whether there are safety issues with the use of some bacterial strains in industrial applications should be discussed.
1. There are multiple problems with references, firstly all species names need to be italicized, DOI is non-required and page number information is incomplete.
2. Version numbers of software such as BlastKoala, KEGG, and ANI should be provided.
3. lines 271-272, the number of the indicator bacterium should be shown.
4. Line 128, L should be L.
5. The manuscript needs to be supplemented with a clear concluding section.
It is OK.
Author Response
Dear Reviewer,
Thank you for the detailed analysis of the manuscript. We took into account your comments and hope that they helped to significantly improve the article.
Overall, this study provides valuable insights into the taxonomic properties, genomic characteristics and biotechnological potential of two bacterial strains isolated from poultry farms in Belgorod, Russia. The results may be useful for further research on these strains or for the development of new applications in the field of microbiology.
Thank you for the for evaluating our work
However, this manuscript has an obvious flaw in that the title "Whole-Genome Sequencing" is not reflected in the manuscript. Since this is a genome-related article, basic information such as genome maps should be displayed.
Done
In addition, genome sequencing-related information should be accessible to the public, e.g. deposited in the NCBI genbank, and published in accession No.
Done
The strains were deposited in the Genbank database under the following numbers:
1.1. – Bioproject PRJNA951603, Biosample SAMN34045332, Genbank acc number JARTEM000000000
1.2. – Bioproject PRJNA951604, Biosample SAMN34045556, Genbank acc number JARTEN000000000
With regard to the biotechnological potential of these strains, it is not clear whether these experiments were performed under optimal conditions or whether other factors could have influenced the results. Whether there are safety issues with the use of some bacterial strains in industrial applications should be discussed.
Done. Line 411
- There are multiple problems with references, firstly all species names need to be italicized, DOI is non-required and page number information is incomplete.
Done. Line 436
- Version numbers of software such as BlastKoala, KEGG, and ANI should be provided.
Done
- lines 271-272, the number of the indicator bacterium should be shown.
The text presents data from a source whose authors did not provide the numbers of indicator strains, therefore, based on the source, it is impossible to indicate this information
- Line 128, L should be L.
L and D in the name of the substance are not italicized. However, if the editor recommends highlighting, we will.
- The manuscript needs to be supplemented with a clear concluding section.
Done. Line 397
Comments on the Quality of English Language
It is OK.
Thank you once again for your comments, which allowed us to correct the manuscript.
On behalf of all co-authors, Nikita Lyakhovchenko.

Reviewer 2 Report
The study deals about an isolation project from poultry waste: two relevant strains were detected, Gram-positive bacteria, bacilliform in aspect and facultatively anaerobic bacteria. It coulf be useful to add some more description for tables
Good exposition
Author Response
Dear Reviewer,
Thank you for your comments. On behalf of all co-authors, Nikita Lyakhovchenko.

Reviewer 3 Report
Dear Authors,
This is an interesitng paper howver, the follwoing must be changed:
Abstract: Reduce the abstract to 200 words maximum as requested by "Instruction for authors".
Line 58: What is this parenthesis? Please add a citation.
Line 60: Do you mean 180 g of manure per chicken per day? Clarify
Line 65: E. coli group bacteria? You mean Enterobacteriaceae? Rephrase
Line 173: Lysis zone? You mean growth inhibition zone?
Line 173: You need to explain that for this bacteria species there are no determined cut-offs for resistance profiles.
Line 163: Passivation? You mean passage?
Line 167: Disc diffusion method.
Line 244: You have not uploaded whole genome at Gen Bank? No Ascension numbers? It is important that you do. You say that they were deposited at the All-Russian collection. Please add a citation and a relevant link, I was not able to have access.
The paper need to be edited by a native English-speaker.
Author Response
Dear Reviewer,
Thank you for the detailed analysis of the manuscript. We took into account your comments and hope that they helped to significantly improve the article.
This is an interesitng paper (Thank you for the for evaluating our work
) howver, the follwoing must be changed:
Abstract: Reduce the abstract to 200 words maximum as requested by "Instruction for authors".
Done
Line 58: What is this parenthesis? Please add a citation.
The number of the Federal Law of the authors' country, in which the term "poultry waste" was approved, was indicated in brackets. This would be relevant in the case of publication in the publications of our country, but since we claim to be an international journal, there is no need for this link.
Line 60: Do you mean 180 g of manure per chicken per day? Clarify
Done. Line 47
Line 65: E. coli group bacteria? You mean Enterobacteriaceae? Rephrase
Done. Line 51
Line 173: Lysis zone? You mean growth inhibition zone?
Done. Line 161
Line 173: You need to explain that for this bacteria species there are no determined cut-offs for resistance profiles.
Done. Line 409
Line 163: Passivation? You mean passage?
Done. Line 151
Line 167: Disc diffusion method.
Done. Line 155
Line 244: You have not uploaded whole genome at Gen Bank? No Ascension numbers? It is important that you do. You say that they were deposited at the All-Russian collection. Please add a citation and a relevant link, I was not able to have access.
Done. Line 224. The strains were deposited in the Genbank database under the following numbers:
1.1. – Bioproject PRJNA951603, Biosample SAMN34045332, Genbank acc number JARTEM000000000
1.2. – Bioproject PRJNA951604, Biosample SAMN34045556, Genbank acc number JARTEN000000000
Comments on the Quality of English Language
The paper need to be edited by a native English-speaker.
Done.
Thank you once again for your comments, which allowed us to correct the manuscript.
On behalf of all co-authors, Nikita Lyakhovchenko.

Round 2
Reviewer 1 Report
The quality of the revised manuscript has significantly improved. The previously identified deficiencies have been adequately addressed. Additionally, the attention to detail is commendable. I recommend accepting it for publication.
It is ok.